# A Competitive Analysis of Online Failure-Aware Assignment

**Mengjing Chen**[1]    **Pingzhong Tang**[*1]    **Zihe Wang**[*2,3]    **Shenke Xiao**[1]    **Xiwang Yang**[4]

[1]Institute for Interdisciplinary Information Sciences, Tsinghua University, Beijing, China
[2]Gaoling School of Artificial Intelligence, Renmin University of China , Beijing, China
[3]Beijing Key Laboratory of Big Data Management and Analysis Methods, Beijing, China
[4]Bytedance, Beijing, China

## Abstract

Motivated by a new generation of Internet advertising that has emerged in the live streaming e-commerce markets (e.g., Tiktok) over the past five years, we study a variant of online bipartite matching problem: advertisers send ad requests to influencers (aka, key opinion leaders) on a social media platform. Each influencer has a maximum number of ad requests she can accommodate. We assign a fixed number of influencers to an advertiser when she enters the platform. The advertiser then matches with each of the assigned influencers with a probability, which can be thought of as a set of negotiations between the advertiser and the set of assigned influencers. Unlike the standard online assignment problems, the outcome of any of these matches is not revealed throughout the session (negotiations take time). Our goal is to maximize the expected number of matches between advertisers and influencers.

We put forward a new deterministic algorithm with a competitive ratio of $1/2$ and prove that no deterministic algorithm can achieve a better competitive ratio. We also show that the competitive ratio can be improved when randomness is allowed. We then study a setting where a match is successful with either probability 0 or a fixed $p$. We present an optimal randomized algorithm that achieves a competitive ratio of $1 - 1/e$ in this setting.

## 1 INTRODUCTION

Live streaming e-commerce promotes and sells products through live webcasts on social media platforms. Over the past few years, such live streaming e-commerce markets have grown fiercely. To put the numbers in perspective, in China of the year 2019 alone, the total Gross Mechanize Volume (GMV) of transactions in live streaming e-commerce is over \$63 billion, and the GMV has just doubled in 2020 Ma [2020]. In these markets, advertisers seek online *influencers* to present their products on their live webcasts. Similar to traditional TV shopping channels, such demonstrations are more vivid compared to those text and picture ads. Unlike the TV shopping channels, such ads are personalized, displaying only to a group of buyers known to have higher conversion rates. It is reported that the conversion rate of such advertisements is at least $21.1\%$ while that of an ordinary ad is less than $5\%$ Grazian [2019], iResearch [2019].

Advertising is an important way for influencers to monetize their fame and traffic, so social media giants such as TikTok and Instagram provide a matching market that facilitates the cooperation between advertisers and influencers. When an advertiser enters such a matching market, she can see the information of all available influencers on the platform and send advertisement requests to them. Despite having full information of influencers, it is difficult for an advertiser to find the most suitable one. On the one hand, the significant number of influencers makes it impossible for the advertiser to look through all the information. On the other hand, a famous influencer can receive many advertisement requests and reject some due to time capacity. To optimize successful matches between advertisers and influencers, it needs a centralized matching algorithm that recommends several selected influencers for each advertiser once she appears online, without knowing any information about future advertisers. The advertiser then negotiates with each of these influencers and matches with a given probability (the outcome of whether any of these matches is successful will reveal only after the whole assignment is made). In this paper, we model this problem as a new variant of the online assignment problem and provide algorithmic solutions. The platform's objective is to maximize the expected number of successful matches, with each influencer having its service capacity.

The problem we focus on can be formulated as the online

---

*corresponding author

assignment problem with stochastic rewards in the setting where the outcome of any match is not revealed before the whole assignment process ends. We call this assignment problem the *failure-aware assignment problem*. Despite the vast literature on online bipartite matching and assignment problems in the past Karp et al. [1990], Goel and Mehta [2008], Feldman et al. [2009, 2010], this problem has not been investigated to the best of our knowledge.

To compare approaches, we define the competitive ratio of an algorithm to be the ratio of the objective value produced by the algorithm to that produced by an optimal solution for the *worst-case*. In Section 3, we propose a greedy deterministic algorithm whose competitive ratio is at least $1/2$ and prove that no deterministic algorithm can achieve a better competitive ratio. This implies that our algorithm is optimal among all deterministic algorithms.

In Section 4, we show that the randomized algorithms can achieve higher competitive ratios even in constrained cases, and we propose a randomized algorithm with a tight competitive ratio in such cases. This is a theoretical improvement to our greedy deterministic algorithm proposed in Section 3. We study the setting where the probabilities of success transactions are either 0 or a fixed value $p$ in Section 5. We propose an optimal randomized algorithm that achieves a competitive ratio of $1 - 1/e$ in this setting.

## 1.1 RELATED WORKS

Our work is closely related to the *AdWords problem* Mehta et al. [2005], Devanur and Hayes [2009], Buchbinder et al. [2007], which is a generalization of the online bipartite matching problem. In the AdWords problem, an Internet search engine company selects an advertisement to display when each query comes, given the advertisers' budgets and bids. The company's goal is to design the allocation rule to maximize its revenue. The AdWords model looks similar to our model where only one influencer is recommended to an advertiser. However, the objective values are evaluated differently by the two problems. For example, if 1 advertiser with budget 1 is assigned to 2 queries with both bids 0.5, then the advertiser contributes 1 to the revenue of the company, but it only contributes $(1 - 0.5)^2 \times 0 + \left(1 - (1 - 0.5)^2\right) \times 1 = 0.75$ to the target value of our problem (see Section 2). The AdWords problem can be formulated as a linear program while our problem cannot. This difference makes the methods solving the AdWords problem Mehta et al. [2005], Buchbinder et al. [2007] not suitable for our problem.

The AdWords problem, as well as the online bipartite matching problem, is a special case of the online submodular welfare maximization problem Nemhauser et al. [1978], Fisher et al., Nemhauser and Wolsey [1978], Kapralov et al. where the objective function is budget-additive. The objec-

tive function is indeed submodular in our problem but not budget-additive. In the classical submodular welfare maximization problem, each item (advertiser) is only allowed to be allocated to one agent (influencer). We explore the more general setting where multiple influencers are recommended to each advertiser in our problem. Moreover, we study the worst-case competitive ratios of randomized algorithms.

Another similar setting is studied in the literature Mehta et al. [2014], Mehta and Panigrahi [2012], Goyal and Udwani [2020]. They realize the problem of the assignment failure and initiate the online stochastic assignment problem. There are some fundamental differences: (1) They focus on the matching problem where each node cannot match with more than one other node. At the same time, either the advertisers or the influencers can have multiple cooperators in our setting. The matching problem is a special case of our problem where each influencer's capacity is 1and the platform only recommends one available influencer to each advertiser. (2) In their setting, the outcomes of the success of matches of previous nodes are known when a new node arrives, while in our setting, we never know the realization from beginning to end.

## 2 PROBLEM FORMULATION

Because our model can be widely used in many scenarios, we will strip the Internet-advertising background from our model and describe it as an assigning-node-to-arrival problem as follows. There are $n$ nodes (corresponding to influencers) and $m$ arrivals (corresponding to advertisers). These arrivals arrive one by one. For ease of representation, we number the arrivals $1, 2, \ldots, m$ by the order they arrive, that is, arrival 2 arrives after arrival 1, arrival 3 arrives after arrival 2, etc. When arrival $i$ arrives, we are required to immediately assign $s$ different nodes to it. For each arrival $i$ and each node $j$, if we assign node $j$ to arrival $i$, arrival $i$ will *accept* node $j$ with probability $p_{ij}$. Whether an arrival accepts a node is independent of each other and remains unknown to us along the whole process. The probabilities $p_{i1}, p_{i2}, \ldots, p_{in}$ are revealed to us immediately after arrival $i$ arrives. Moreover, each node $j$ has a capacity $c_j$, meaning the maximum number of arrivals to which it is able to be *successfully* assigned. More precisely, let $P_{ij}$ denote a random variable whose value is 1 with probability $p_{ij}$ and 0 with probability $1 - p_{ij}$ (these random variables are mutually independent), then for node $j$, if we assign it to arrivals $i_1, i_2, \ldots, i_k$, it will be *successfully* assigned to $\mathbb{E}\left(\min\{P_{i_1 j} + \cdots + P_{i_k j}, c_j\}\right)$ arrivals in expectation[1]. Our target is to maximize the expected number of successful assignments, or formally, to solve the following program in an online fashion.

---

[1]Since we only care the expected number of successful assignments, it doesn't matter to which arrivals this node is successfully assigned.

$$\max \quad \sum_j \mathbb{E}\left(\min\left\{\sum_i x_{ij}P_{ij}, c_j\right\}\right)$$

$$\text{s.t.} \quad \sum_j x_{ij} = s, \qquad\qquad \text{for all } i,$$

$$x_{ij} \in \{0, 1\}, \qquad\qquad \text{for all } i, j.$$

In the online setting, the values of $x_{1j}$'s, $x_{2j}$'s, ... are determined in order, and when determining the value of $x_{ij}$, we don't know the values of $p_{i'j'}$'s for $i' > i$.

**Remark** Note that assigning a node to arrival is always no worse than not assigning it whenever an arrival arrives, so we assume exactly $s$ nodes are assigned to the arrival in the setting instead of no more than $s$ nodes. This also implies $s \leq n$. If $s > n$, we can add some hypothetical nodes, which are not accepted by any arrival (i.e., accepted with probability 0), to make $n \geq s$.

**Example 1.** *Consider an instance with 2 arrivals and 2 nodes, and $s = 1$, i.e., each time an arrival arrives, we only assign one node to it. In this instance, $c_1 = c_2 = 1$, $p_{11} = p_{12} = p_{21} = 0.5$ and $p_{22} = 0$. Suppose an algorithm $\mathcal{A}$ assigns node 1 to arrival 1. When arrival 2 arrives, since $p_{22} = 1$, it is optimal for $\mathcal{A}$ to assign node 1 to arrival 2. The objective value produced by $\mathcal{A}$ is exactly $\mathbb{E}(\min\{P_{11} + P_{21}, 1\}) = 0.75$ (recall that $P_{11}$ and $P_{12}$ are i.i.d. random variables which take value 1 with probability 0.5 and 0 with probability 0.5). Note the optimal assignment for this example would assign node 2 to arrival 1 and node 1 to arrival 2, and the optimal objective value is $\mathbb{E}(\min\{P_{12}, 1\}) + \mathbb{E}(\min\{P_{21}, 1\}) = 1$.*

**Measurement** Given an algorithm $\mathcal{A}$ and an instance ins of this problem, we define $\mathcal{A}(\text{ins})$ as the expected objective value produced by $\mathcal{A}$, i.e., the expected value of $\sum_j \mathbb{E}(\min\{\sum_i x_{ij}P_{ij}, c_j\})$ where $x_{ij}$'s are outputted by $\mathcal{A}$ when running on the instance ins. Here we say "expected value" because $\mathcal{A}$ may be a randomized algorithm. Furthermore, we define the *competitive ratio* of an algorithm $\mathcal{A}$ as:

$$\inf_{\text{ins}} \frac{\mathcal{A}(\text{ins})}{\max_{\mathcal{A}'} \mathcal{A}'(\text{ins})}.$$

The competitive ratio of an algorithm is the ratio of the expected objective value produced by the algorithm to that produced by an optimal assignment under the *worst-case* instance.

## 3 DETERMINISTIC ALGORITHM

In this section, we put our attention to deterministic algorithms. We first propose a deterministic algorithm with a competitive ratio of $1/2$; we prove $1/2$ is the upper bound

of the competitive ratio that deterministic algorithms can achieve.

Our algorithm is a greedy algorithm whose greedy policy is to assign nodes to increase the current objective value as much as possible whenever an arrival arrives. Formally, we define

$$w_{ij} = \mathbb{E}\left(\min\left\{P_{ij} + \sum_{i':i'<i} x_{i'j}P_{i'j}, c_j\right\}\right)$$
$$- \mathbb{E}\left(\min\left\{\sum_{i':i'<i} x_{i'j}P_{i'j}, c_j\right\}\right).$$

The greedy algorithm assigns $x_{ij_1}, \ldots, x_{ij_s}$ to 1 (and assigns $x_{ij'}$'s to 0 for $j' \notin \{j_1, \ldots, j_s\}$) where $j_1, \ldots, j_s$ are indices that maximize $\sum_{k=1}^s w_{ij_k}$.

To simplify the representation, we denote by $p_t$ the probability that the random variable $\min\{\sum_{i':i'<i} x_{i'j}P_{i'j}, c_j\}$ takes value $t$, then we have

$$\mathbb{E}\left(\min\left\{P_{ij} + \sum_{i':i'<i} x_{i'j}P_{i'j}, c_j\right\}\right)$$
$$= \sum_{t=1}^{c_j-1} t\left(p_t(1-p_{ij}) + p_{t-1}p_{ij}\right) + c_j\left(p_{c_j} + p_{c_j-1}p_{ij}\right)$$
$$= \sum_{t=1}^{c_j} tp_t + p_{ij}\left(1 - p_{c_j}\right)$$
$$= \mathbb{E}\left(\min\left\{\sum_{i':i'<i} x_{i'j}P_{i'j}, c_j\right\}\right) + p_{ij}\left(1 - p_{c_j}\right),$$

i.e.,

$$w_{ij} = p_{ij}\left(1 - p_{c_j}\right). \tag{1}$$

Hence, we can maintain the distribution of $\min\{\sum_{i':i'<i} x_{i'j}P_{i'j}, c_j\}$ so that for each arrival, the algorithm takes $O(n\log n)$ time to find the nodes $j_1, \ldots, j_s$ to assign plus $O(\sum_{k=1}^s c_{j_s})$ time to update the distribution of $\min\{\sum_{i':i'<i} x_{i'j}P_{i'j}, c_j\}$. We call this algorithm GREEDY.

**Theorem 1.** *GREEDY has a competitive ratio of at least $1/2$.*

*Proof.* Suppose when following the assignment produced by GREEDY, node $j$ is successfully assigned to $\alpha_j c_j$ arrivals in expectation, and when following an optimal assignment, node $j$ is successfully assigned to $b_j$ arrivals in expectation. Let ALG be the objective value when following the assignment produced by GREEDY, and let OPT be the objective value when following the optimal assignment. We have immediately $\text{OPT} = \sum_j b_j$ and

$$\text{ALG} = \sum_j \alpha_j c_j \geq \sum_j \alpha_j b_j. \tag{2}$$

On the other hand, suppose for arrival $i$, the optimal solution assigns nodes $j_{i1}, \ldots, j_{is}$ while GREEDY assigns nodes $j'_{i1}, \ldots, j'_{is}$. By the greedy policy of GREEDY, we have (in this proof, the variables $x_{ij}$'s and $w_{ij}$'s refer to the ones produced by GREEDY)

$$\sum_{k=1}^{s} w_{ij'_{ik}} \geq \sum_{k=1}^{s} w_{ij_{ik}} \tag{3}$$

Now let us fix a value $k \in \{1, \ldots, s\}$, and suppose $\min\left\{\sum_{i':i'<i} x_{i'j_{ik}} P_{i'j_{ik}}, c_{j_{ik}}\right\}$ takes value $k$ with probability $p_k$, then by (1) we have

$$w_{ij_{ik}} = p_{ij_{ik}}\left(1 - p_{c_{j_{ik}}}\right) \geq p_{ij_{ik}}\left(1 - \alpha_{j_{ik}}\right) \tag{4}$$

where the last inequality holds because

$$\alpha_{j_{ik}} = \frac{\mathbb{E}\left(\min\left\{\sum_{i'} x_{i'j_{ik}} P_{i'j_{ik}}, c_{j_{ik}}\right\}\right)}{c_{j_{ik}}}$$
$$\geq \frac{\mathbb{E}\left(\min\left\{\sum_{i':i'<i} x_{i'j_{ik}} P_{i'j_{ik}}, c_j\right\}\right)}{c_{j_{ik}}}$$
$$\geq \frac{p_{c_{j_{ik}}} c_{j_{ik}}}{c_{j_{ik}}} = p_{c_{j_{ik}}}.$$

By summing up (4) for $k$ from 1 to $s$, we have

$$\sum_{k=1}^{s} w_{ij_{ik}} \geq \sum_{k=1}^{s} p_{ij_{ik}}\left(1 - \alpha_{j_{ik}}\right). \tag{5}$$

Therefore,

$$\text{ALG} = \sum_i \sum_{k=1}^{s} w_{ij'_{ik}} \geq \sum_i \sum_{k=1}^{s} w_{ij_{ik}} \quad \text{(by (3))}$$
$$\geq \sum_i \sum_{k=1}^{s} p_{ij_{ik}}\left(1 - \alpha_{j_{ik}}\right) \quad \text{(by (5))}$$
$$= \sum_j \sum_{(i,k):j_{ik}=j} (1 - \alpha_j) p_{ij} \geq \sum_j (1 - \alpha_j) b_j.$$

Combined with (2), we have

$$\text{ALG} \geq \frac{1}{2}\left(\sum_j \alpha_j b_j + \sum_j (1 - \alpha_j) b_j\right)$$
$$= \frac{1}{2} \sum_j b_j = \frac{1}{2}\text{OPT}.$$

Note the argument above works for any instance of the problem, so the competitive ratio of GREEDY is at least 1/2. □

The following theorem shows that no deterministic algorithm can achieve a competitive ratio better than 1/2, meaning that GREEDY is optimal among all deterministic algorithms in the sense of competitive ratio.

**Theorem 2.** *For any deterministic algorithm $\mathcal{A}$ for our problem, the competitive ratio of $\mathcal{A}$ is no more than $1/2$ even if there are only 2 arrivals and only 1 node is allowed to be assigned to each arrival, i.e.,*

$$\inf_{\text{ins}} \frac{\mathcal{A}(\text{ins})}{\max_{\mathcal{A}'} \mathcal{A}'(\text{ins})} \leq \inf_{\text{ins}:n\leq 2,s=1} \frac{\mathcal{A}(\text{ins})}{\max_{\mathcal{A}'} \mathcal{A}'(\text{ins})} \leq \tfrac{1}{2}.$$

*Proof.* We construct an instance $I_1$ with 2 arrivals and 2 nodes, and let $c_1 = c_2 = 1$, $p_{11} = p_{12} = p_{21} = 1$, and $p_{22} = 0$. In addition, we construct another instance $I_2$ that is almost the same as $I_1$ except that $p_{21} = 0$ and $p_{22} = 1$. For $I_1$, an algorithm can assign node 2 to arrival 1 and assign node 1 to arrival 2 to achieve an objective value 2. For $I_2$, the objective value 2 can also be achieved by assigning node 1 to arrival 1 and assigning node 2 to arrival 2. Hence, we have $\max_{\mathcal{A}'} \mathcal{A}'(I_1) \geq 2$ and $\max_{\mathcal{A}'} \mathcal{A}'(I_2) \geq 2$.

Now we compare the behavior of $\mathcal{A}$ when running on $I_1$ and $I_2$ respectively. Note when dealing with arrival 1, the information given to $\mathcal{A}$ is the same, and since $\mathcal{A}$ is a deterministic algorithm, it must assign the same node to arrival 1. If $\mathcal{A}$ assigns node 1 to arrival 1, then on instance $I_1$, no arrival will accept node 2 (since $p_{22} = 0$), thus $\mathcal{A}(I_1) = 1$. Similarly, if $\mathcal{A}$ assigns node 2 to arrival 1, then no arrival will accept node 1 on instance $I_2$, thus $\mathcal{A}(I_2) = 1$. Then we have

$$\inf_{\text{ins}\in\{I_1,I_2\}} \frac{\mathcal{A}(\text{ins})}{\max_{\mathcal{A}'} \mathcal{A}'(\text{ins})} \leq \tfrac{1}{2},$$

so the competitive ratio of $\mathcal{A}$ is at most $1/2$. □

## 4 RANDOMIZED ALGORITHM

We have proven the optimal competitive ratio of deterministic algorithms is $1/2$. One may ask whether higher competitive ratios can be achieved if randomized algorithms are allowed. The answer is yes. In this section, we will see that randomness helps improve the competitive ratio even in the very constrained case where $s = 1$.

Recall that our optimal deterministic algorithm GREEDY greedily assigns the node $j$ that maximizes $w_{ij}$ whenever an arrival $i$ arrives. Our randomized algorithm would consider $w_{ij}$'s as the weights and randomly assign a node according to these weights. Also by observing that when arrival $i$ comes, there is no benefit to assign a node whose $w_{ij}$ is not the largest $\min\{m - i + 1, n\}$ ones (for example, it is always optimal to assign the node with the largest $w_{ij}$ to the last arrival), our randomized algorithm only chooses the node from those whose $w_{ij}$'s are the largest $\min\{m - i + 1, n\}$ ones. We call this algorithm RANDOM, which is formally described in Algorithm 1. We will show that RANDOM achieves a higher competitive ratio, which is also the optimal randomized algorithm in this case.

The following theorem shows that the competitive ratio of RANDOM is at least $3/4$ if there are no more than 2

**Algorithm 1** RANDOM

---

When arrival $i$ arrives,

1. For all $j$, let

$$w_{ij} = \mathbb{E}\left(\min\left\{P_{ij} + \sum_{i':i'<i} x_{i'j}P_{i'j}, c_j\right\}\right)$$
$$- \mathbb{E}\left(\min\left\{\sum_{i':i'<i} x_{i'j}P_{i'j}, c_j\right\}\right).$$

   Like the deterministic case, $w_{ij}$'s can be computed efficiently by maintaining the distribution of $\min\left\{\sum_{i':i'<i} x_{i'j}P_{i'j}, c_j\right\}$.

2. Let $k = \min\{m-i+1, n\}$, and find the $k$ largest $w_{ij}$'s: $w_{ij_1}, w_{ij_2}, \ldots, w_{ij_k}$.

3. Assign node $j_t$ with probability $w_{ij_t}/(w_{ij_1} + w_{ij_2} + \cdots + w_{ij_k})$ to arrival $i$.

---

arrivals and only 1 node is allowed to be assigned to each arrival. Note that though the sketch of RANDOM is similar to the GREEDY, the proof techniques are pretty different. Compared with Theorem 2, RANDOM indeed improves the competitive ratio via randomness.

**Theorem 3.** *RANDOM has a competitive ratio of at least $3/4$ if there are no more than 2 arrivals and only 1 node is allowed to be assigned to each arrival.*

$$\inf_{\text{ins}:n\leq 2, s=1} \frac{\text{RANDOM(ins)}}{\max_{\mathcal{A}'} \mathcal{A}'(\text{ins})} \geq \frac{3}{4}.$$

*Proof.* In this proof, we will compare the assignment produced by RANDOM with an optimal assignment. To avoid confusion, we use $x_{ij}$'s to refer to the ones produced by the optimal assignment, while we use $\bar{x}_{ij}$'s to refer to the ones produced by RANDOM (so $\bar{x}_{ij}$'s are random variables). We assume the optimal assignment assigns node $j_i$ to arrival $i$ while RANDOM assigns node $\bar{j}_i$ to arrival $i$. Note that $j_i$ and $\bar{j}_i$ are respectively functions of $x_{i1}, x_{i2}, \ldots, x_{in}$ and $\bar{x}_{i1}, \bar{x}_{i2}, \ldots, \bar{x}_{in}$, thus $\bar{j}_i$ is also a random variable. We define

$$\text{OPT}_i = \mathbb{E}_X\left(\min\left\{P_{ij_i} + \sum_{i':i'<i} x_{i'j_i}P_{i'j_i}, c_{j_i}\right\}\right)$$
$$- \mathbb{E}_X\left(\min\left\{\sum_{i':i'<i} x_{i'j_i}P_{i'j_i}, c_{j_i}\right\}\right)$$

and

$$\text{ALG}_i = \mathbb{E}_X\left(\min\left\{P_{i\bar{j}_i} + \sum_{i':i'<i} \bar{x}_{i'\bar{j}_i}P_{i'\bar{j}_i}, c_{\bar{j}_i}\right\}\right)$$
$$- \mathbb{E}_X\left(\min\left\{\sum_{i':i'<i} \bar{x}_{i'\bar{j}_i}P_{i'\bar{j}_i}, c_{\bar{j}_i}\right\}\right)$$

where $\mathbb{E}_X$ means the expectation is taken over all $P_{i'j}$'s, thus $\text{ALG}_i$ is a random variable. Note by (1) we have

$$\text{OPT}_1 = p_{1j_1}, \tag{6}$$
$$\text{ALG}_1 = p_{1\bar{j}_1}, \tag{7}$$
$$\text{OPT}_2 = p_{2j_2}\left(1 - x_{1j_2}[c_{j_2} \leq 1]p_{1j_2}\right), \tag{8}$$

where [condition] is an indicator that equals to 1 if the condition is true and 0 otherwise. Now the competitive ratio of the randomized algorithm can be expressed as

$$\min \frac{\sum_i \mathbb{E}(\text{ALG}_i)}{\sum_i \text{OPT}_i},$$

where the minimum is taken over instances.

We first analyze $\text{ALG}_1$. Assume $p_{1\ell_1}$ and $p_{1\ell_2}$ are the largest 2 ones among all $p_{1j}$'s. Recall that when RANDOM deals with arrival 1, $w_{1j}$'s are exactly $p_{1j}$'s, so

$$\mathbb{E}(\text{ALG}_1) = \mathbb{E}\left(p_{1\bar{j}_1}\right) \qquad \text{(by (7))}$$
$$= \frac{p_{1\ell_1}}{p_{1\ell_1} + p_{1\ell_2}} \cdot p_{1\ell_1} + \frac{p_{1\ell_2}}{p_{1\ell_1} + p_{1\ell_2}} \cdot p_{1\ell_2} \tag{9}$$
$$= \frac{1 + (p_{1\ell_2}/p_{1\ell_1})^2}{1 + p_{1\ell_2}/p_{1\ell_1}} \cdot p_{1\ell_1}$$
$$\geq 2\left(\sqrt{2} - 1\right)\text{OPT}_1, \tag{10}$$

where the inequality (10) holds by (6) and taking the minimum of the function $\left(1 + t^2\right)/(1 + t)$ over $[0, 1]$.

We then analyze $\text{ALG}_2$. According to the rule of RANDOM, since arrival 2 is the last arrival, the algorithm will deterministically assign node $j$ that maximizes $w_{2j}$, i.e.,

$$\text{ALG}_2 \geq w_{2j_2} = p_{2j_2}\left(1 - \bar{x}_{1j_2}[c_{j_2} \leq 1]p_{1j_2}\right). \quad \text{(by (1))}$$

Hence,

$$\mathbb{E}(\text{ALG}_2) \geq p_{2j_2}\left(1 - \mathbb{E}(\bar{x}_{1j_2})[c_{j_2} \leq 1]p_{1j_2}\right). \tag{11}$$

After comparing (11) with (8), we can see if $\mathbb{E}(\bar{x}_{1j_2}) \leq x_{1j_2}$, we have $\mathbb{E}(\text{ALG}_2) \geq \text{OPT}_2$, thus

$$\mathbb{E}(\text{ALG}_1) + \mathbb{E}(\text{ALG}_2) \geq 2\left(\sqrt{2} - 1\right)\text{OPT}_1 + \text{OPT}_2$$
$$\geq 3(\text{OPT}_1 + \text{OPT}_2)/4,$$

which completes the proof. Hence, in the rest of the proof, we assume $\mathbb{E}(\bar{x}_{1j_2}) > x_{1j_2}$. With this assumption, we can assert that RANDOM has a non-zero probability to assign node $j_2$ to arrival 1, which means $j_2 \in \{\ell_1, \ell_2\}$ by the rules of RANDOM, and the optimal assignment does not assign node $j_2$ to arrival 1, which means $j_1 \neq j_2$. Furthermore, we can assume $j_1 \in \{\ell_1, \ell_2\}$, otherwise we can change $j_1$ to an index in $\{\ell_1, \ell_2\} \setminus \{j_2\}$, which does not reduce the target value of the optimal solution. As a result, we have $\{j_1, j_2\} = \{\ell_1, \ell_2\}$. Hence, we can rewrite (9) as

$$\mathbb{E}(\text{ALG}_1) \geq \frac{p_{1j_1}^2 + p_{1j_2}^2}{p_{1j_1} + p_{1j_2}}. \tag{12}$$

Also, according to the rules of RANDOM, node $j_2$ is assigned to arrival 1 with probability $p_{1j_2}/(p_{1j_1} + p_{1j_2})$, so $\mathbb{E}(\bar{x}_{1j_2}) = p_{1j_2}/(p_{1j_1} + p_{1j_2})$, and

$$\mathbb{E}(\text{ALG}_2) \geq \text{OPT}_2 \left(1 - \frac{p_{1j_2}^2}{p_{1j_1} + p_{1j_2}}\right). \qquad (13)$$

By combining (6), (10) and (13), we have

$$\frac{\mathbb{E}(\text{ALG}_1) + \mathbb{E}(\text{ALG}_2)}{\text{OPT}_1 + \text{OPT}_2}$$

$$\geq \frac{\frac{p_{1j_1}^2 + p_{1j_2}^2}{p_{1j_1} + p_{1j_2}} + \text{OPT}_2 \left(1 - \frac{p_{1j_2}^2}{p_{1j_1} + p_{1j_2}}\right)}{p_{1j_1} + \text{OPT}_2}$$

$$\geq \min \left\{ \frac{p_{1j_1}^2 + p_{1j_2}^2}{p_{1j_1}(p_{1j_1} + p_{1j_2})}, \frac{\frac{p_{1j_1}^2}{p_{1j_1} + p_{1j_2}} + 1}{p_{1j_1} + 1} \right\} \qquad (14)$$

$$\geq \frac{3}{4}. \qquad (15)$$

Here the inequality (14) uses the fact that $(ka + b)/(a + c) \geq \min\{b/c, (k+b)/(1+c)\}$ for $0 \leq a, b, c \leq 1$, and the inequality (15) holds because it is equivalent to two inequalities corresponding to the two parts of "min", and each inequality can be turned into a quadratic inequality, which is easy to validate. $\qquad \square$

The competitive ratio of $3/4$ is tight. We formalize it as the following theorem.

**Theorem 4.** *For any (randomized) algorithm $\mathcal{A}$ for our problem, the competitive ratio of $\mathcal{A}$ is no more than $3/4$ even if there are no more than 2 arrivals and only 1 node is allowed to be assigned to each arrival, i.e.,*

$$\inf_{\text{ins}} \frac{\mathcal{A}(\text{ins})}{\max_{\mathcal{A}'} \mathcal{A}'(\text{ins})} \leq \inf_{\text{ins}: n \leq 2, s=1} \frac{\mathcal{A}(\text{ins})}{\max_{\mathcal{A}'} \mathcal{A}'(\text{ins})} \leq \frac{3}{4}.$$

*Proof.* By Yao's lemma Yao [1977], we only need to consider deterministic algorithms on randomized inputs. We construct a randomized instance $I$ with 2 arrivals and 2 nodes, and let $c_1 = c_2 = 1$, $p_{11} = p_{12}$. Moreover, we set $p_{21} = 1, p_{22} = 0$ with probability $1/2$ and $p_{21} = 0, p_{22} = 1$ with probability $1/2$. Now consider the best deterministic algorithm on this randomized instance. No matter which node the algorithm assigns to arrival 1, the expected competitive ratio is $(1 + 1/2)/2 = 3/4$, so the competitive ratio of any (randomized) algorithm cannot exceed $3/4$. $\qquad \square$

Unfortunately, RANDOM may perform asymptotically bad when the number of arrivals increases. Consider an instance where $p_{11} = 1, p_{12} = \cdots = p_{1n} = \epsilon$, and $p_{ij} = 0$ for all $i \geq 2$ and all $j$. An optimal assignment would assign node 1 to arrival 1, which obtains a target value of 1. However, RANDOM will produce an expected target

value of $\left(1 + (n-1)\epsilon^2\right)/(1 + (n-1)\epsilon)$. So if we take $\epsilon = 1/\sqrt{n-1}$, the expected target value produced by RANDOM will converge to 0 when $n$ tends to infinity.

## 5 RANKING ALGORITHM

In this section, we consider the case where all $c_j$'s are the same (say $c$), and each $p_{ij}$ is either 0 or a fixed value $p$ ($0 < p \leq 1$). The case can be applied to markets where the differences of effect and fame of influencers are small. The social media platforms that mainly display professional production always have this feature.

For each arrival $i$, we define the *feasible set* for arrival $i$ as $F_i = \{j \mid p_{ij} > 0\}$. In this setting, when arrival $i$ comes, there are possibly multiple $j$'s that maximize $w_{ij}$'s. We show that if we choose the nodes according to an order randomly determined in advance, the competitive ratio can be improved to $1 - 1/e$. The idea of this algorithm comes from the RANKING algorithm in Karp et al. [1990]. We call this algorithm RANKING, which is formally described in Algorithm 2. Note the classical online bipartite matching problem Karp et al. [1990] is exactly a special case in this setting where $p = c = s = 1$. Since Karp et al. have proved that the upper bound for the competitive ratio of the classical online bipartite matching problem is $(1 - 1/e) + o(1)$, our RANKING algorithm is optimal in this setting.

---
**Algorithm 2** RANKING

1. Sort all nodes in a random order.

2. When arrival $i$ arrives,
   
   (a) For all $j$, let $y_{ij} = \sum_{i': i' < i} x_{i'j}$, i.e. the number of arrivals to which node $j$ has already been assigned.
   
   (b) We rename all $j$'s as $j_{i,1}, j_{i,2}, \ldots$ such that $w_{j_{i,1}} = \cdots = w_{j_{i,k_1}} > w_{j_{i,k_1+1}} = \cdots = w_{j_{i,k_2}} > \cdots$, where $j_{i,1} < \cdots < j_{i,k_1}$, $j_{i,k_1+1} < \cdots < j_{i,k_2}$, and so on.
   
   (c) Assign nodes $j_{i,1}, \ldots, j_{i,s}$ to arrival $i$.

---

**Theorem 5.** *In the special case where all $c_j$'s are the same, and each $p_{ij}$ is either 0 or a fixed value $p$ ($0 < p \leq 1$), RANKING has a competitive ratio of $1 - 1/e$.*

The proof is analogous to the one in Karp et al. [1990]. The key idea of their proof is to turn the original setting into a setting where arrivals are known at the beginning while nodes arrive one by one according to the random order generated by RANKING instead. However, in our setting, even when $s = 1$, one node can be assigned to multiple arrivals, so it does not make the problem easier to use this idea directly. We handle this difficulty by allowing nodes to

arrive round after round. This makes our proof much more complicated.

*Proof.* We first prove this algorithm has a competitive ratio of $1 - 1/e$ in the case where $s = 1$.

Given an order $\sigma$ of nodes $\sigma_1, \ldots, \sigma_n$, we consider a setting where arrivals are known at the beginning while nodes arrive one by one instead according to $\sigma$. Specifically, node $\sigma_1$ arrives first, then node $\sigma_2$ arrives, and so on. When a node arrives, we are asked to assign it to an arrival that has not been assigned yet. The process above is repeated $m$ times (we call one complete process a *turn*), so a node can be assigned to multiple arrivals through multiple turns. In this setting, we denote by *time $k + n(t-1)$* the moment where node $\sigma_k$ in the $t$-th turn is being assigned. Particularly, when we say "by time $t$", the assigning behavior happening at time $t$ is not included. We call this new setting $\sigma$-DUAL. For any order $\sigma$, we can see that the optimal assignment in $\sigma$-DUAL is the same as the optimal assignment in our original setting.

Consider the following algorithm named $\sigma$-DUAL-RANKING in the setting $\sigma$-DUAL.

---

**Algorithm 3** $\sigma$-DUAL-RANKING

1. When node $j$ arrives,
   (a) Let $i_0$ be the smallest index in $F_i$ such that no node has been assigned to arrival $i_0$ yet.
   (b) Assign node $j_0$ to arrival $i$. If such $i_0$ does not exist, do nothing.

---

We claim that when the random order drawn by RANKING is $\sigma$, $\sigma$-DUAL-RANKING generates the same assignment as RANKING.

**Lemma 1.** *When the random order drawn by RANKING is $\sigma$, $\sigma$-DUAL-RANKING generates the same assignment as RANKING.*

*Proof.* For ease of presentation, we write "during the running of RANKING in the original setting" as "in RANKING" and write "during the running of $\sigma$-DUAL-RANKING in the new setting $\sigma$-DUAL" as "in $\sigma$-DUAL-RANKING". Observing that if $\sigma$-DUAL-RANKING does not assign node $\sigma_j$ to any arrival in some turn, it will not assign node $\sigma_j$ in subsequent turns. Thus we only need to prove the following proposition.

**Proposition 1.** *For any $t$, at time $t$ in $\sigma$-DUAL-RANKING where node $\sigma_j$ is being assigned,*

1. *if node $\sigma_j$ is assigned to arrival $i$ at time $t$, then RANKING will assign node $\sigma_j$ to arrival $i$ too;*

2. *if node $\sigma_j$ is not assigned to any arrival at time $t$, and has already been assigned to arrivals $i_1, \ldots, i_k$ by time $t$, then RANKING will not assign node $\sigma_j$ to any arrival other than $i_1, \ldots, i_k$.*

We prove this proposition by mathematical induction on $t$. We first consider the first part of Proposition 1, i.e., at time $t$ in $\sigma$-DUAL-RANKING, node $\sigma_j$ is assigned to arrival $i$. We assume by this time node $\sigma_j$ has already been assigned to arrivals $i_1, \ldots, i_k$ (i.e., this is the $(k+1)$-th turn). Consider the moment immediately before arrival $i$ arrives in RANKING. If node $\sigma_j$ has been assigned to an arrival $i'$ ($i' < i$) other than $i_1, \ldots, i_k$, then by the induction hypothesis, arrival $i'$ should not be assigned to by time $t$ in $\sigma$-DUAL-RANKING. But $\sigma$-DUAL-RANKING assigns node $\sigma_j$ to arrival $i$ at time $t$ while $i' < i$, which contradicts to the rule of $\sigma$-DUAL-RANKING. Hence, immediately before arrival $i$ arrives in RANKING, node $\sigma_j$ has been assigned to at most $k$ arrivals.

Now we suppose to the contrary that node $\sigma_j$ is not assigned to arrival $i$ by RANKING, then when arrival $i$ arrives in RANKING, another node $\sigma_{j'}$ must be assigned to it. Let $t' < t$ be a time in $\sigma$-DUAL-RANKING where node $\sigma_{j'}$ is being assigned. Note by time $t'$, node $\sigma_{j'}$ has not been assigned to arrival $i$ (otherwise $\sigma$-DUAL-RANKING cannot assign node $\sigma_j$ to arrival $i$), so node $\sigma_{j'}$ must be assigned to some arrival $i''$ at time $t'$ (otherwise by the induction hypothesis, it will never be assigned to arrival $i$ in RANKING). Since $\sigma$-DUAL-RANKING assigns node $\sigma_{j'}$ to arrival $i''$ rather than $i$, we have $i'' < i$ in addition by the rule of $\sigma$-DUAL-RANKING. Hence, by the induction hypothesis, when arrival $i$ arrives in RANKING, node $\sigma_{j'}$ is assigned to at least $k$ (if $j' > j$) or $k+1$ (if $j' < j$) arrivals. Recall that we have shown that at the same time, node $\sigma_j$ has been assigned to at most $k$ arrivals, so RANKING will choose to assign node $\sigma_j$ to arrival $i$ by its rule, a contradiction, which proves the first part of Proposition 1.

We then consider the second part of Proposition 1, i.e., $\sigma$-DUAL-RANKING, node $\sigma_j$ is not assigned to any arrival at time $t$ and has already been assigned to arrivals $i_1, \ldots, i_k$ by time $t$. Suppose to the contrary that RANKING assigns node $\sigma_j$ to an arrival $i$ other than $i_1, \ldots, i_k$, then at time $t$ in $\sigma$-DUAL-RANKING, some node $\sigma_{j'}$ must have been assigned to arrival $i$, otherwise $\sigma$-DUAL-RANKING will assign node $\sigma_j$ to arrival $i$ (or an arrival $i'$ with $i' < i$) by its rule. By the induction hypothesis, node $\sigma_{j'}$ is also assigned to arrival $i$ in RANKING, a contradiction. $\square$

By Lemma 1, it is sufficient to prove when $\sigma$ is randomly generated, the ratio of the expected target value generated by $\sigma$-DUAL-RANKING to the optimal value (note the optimal value is the same for any $\sigma$) is at least $(1 - 1/e)$ in the setting $\sigma$-DUAL. Next, we focus on the setting $\sigma$-DUAL.

We define $P_1, P_2, \ldots$ to be i.i.d. random variables whose

value is 1 with probability $p$ and 0 with probability $1 - p$. Let $v(k) = \mathbb{E}\left(\min\{P_1 + \cdots + P_k, c\}\right)$ and $d(k) = v(k) - v(k-1)$. Now consider an arbitrary algorithm. For an arrival $i$, if the algorithm assigns node $\sigma_j$ to it at time $t$ in turn $k$, and node $\sigma_j$ has already been assigned to $k'$ ($k' < k$) arrivals by time $t$, then assigning this node to arrival $i$ increases the target value by $d(k'+1)$. We define the *weight* of arrival $i$ to be $d(k'+1)$. In addition, we define the *fake weight* of arrival $i$ to be $d(k)$. If no node is assigned to arrival $i$, both the fake weight and the weight of arrival $i$ are defined to be 0. Note the (fake) weights of arrivals depend on the algorithm, and the sum of the weights of all arrivals is exactly the target value generated by the algorithm.

We call an algorithm a *refusal algorithm* if when a node arrives, it either assigns it to the arrival chosen by the rule of $\sigma$-DUAL-RANKING or does not assign it to any arrival. We define the *fake value* of a refusal algorithm to be the sum of the fake weights of all arrivals.

**Lemma 2.** *The fake value of any refusal algorithm is no more than the target value generated by $\sigma$-DUAL-RANKING.*

*Proof.* We first fix a refusal algorithm $\mathcal{R}$. We respectively denote by $z_i$ and $z_i'$ the weights and fake weights of arrival $i$ corresponding to $\sigma$-DUAL-RANKING and $\mathcal{R}$. Suppose $\sigma$-DUAL-RANKING and $\mathcal{R}$ respectively assign a node to arrival $i$ at time $t_i$ and $t_i'$ (if no node is assigned to arrival $i$, $t_i$ or $t_i'$ is defined to be $+\infty$), we define

$$z_i(t) = \begin{cases} z_i, & \text{if } t > t_i \\ 0, & \text{otherwise} \end{cases}, \quad z_i'(t) = \begin{cases} z_i', & \text{if } t > t_i' \\ 0. & \text{otherwise} \end{cases}$$

Note $z_i(nm + 1) = z_i$ and $z_i'(nm + 1) = z_i'$.

We prove the following stronger proposition instead by mathematical induction on $t$.

**Proposition 2.** *For any $t$, $z_i(t) \geq z_i'(t)$*

It trivially holds for $t = 1$. Consider time $t - 1$ in the $k$-th turn. If $\mathcal{R}$ does nothing at time $t - 1$, then for any $t$, $z_i'(t) = z_i'(t - 1)$ while $z_i(t) \geq z_i(t - 1)$, we have $z_i(t) \geq z_i'(t)$ by the induction hypothesis.

If $\mathcal{R}$ assigns node $\sigma_j$ to arrival $i$ at time $t - 1$, then $z_i'(t) = d(k)$. If by time $t - 1$, $\sigma$-DUAL-RANKING has assigned some nodes to arrival $i$, then $z_i(t) \geq d(k) = z_i'(t)$ (note $d(k)$ is non-increasing in $k$). Otherwise, suppose $\sigma$-DUAL-RANKING assigns node $\sigma_j$ to arrival $i'$ at time $t$. Since $\sigma$-DUAL-RANKING is able to assign node $\sigma_j$ to arrival $i$ at this time, we have $i' \leq i$. If $i' < i$, since $\mathcal{R}$ does not assign node $\sigma_j$ to arrival $i'$, it must have assigned some node to arrival $i'$ by time $t$, which means $z_{i'}'(t) > 0$, thus $z_{i'}(t) > 0$ by the induction hypothesis, i.e., $\sigma$-DUAL-RANKING must also have assigned some node to arrival $i'$ by time $t$, which contradicts to our assumption that $\sigma$-DUAL-RANKING

assigns node $\sigma_j$ to arrival $i'$ at time $t$. If $i' = i$, we have $z_i(t) = z_i'(t)$. Note $z_{i''}(t) = z_{i''}(t - 1)$ and $z_{i''}'(t) = z_{i''}'(t - 1)$ for any arrival $i''$ other than $i$, we have $z_i(t) \geq z_i'(t)$ for any $i$ by the induction hypothesis. $\square$

Suppose algorithm $\mathcal{O}$ generates an optimal assignment. Let $A_k$ be the set of arrivals whose weight is $d(k)$ corresponding to $\mathcal{O}$. Let $\mathcal{R}$ be a refusal algorithm that does not assign any node to an arrival not in $A_k$ in turn $k$. We can see $\mathcal{R}$ sequentially performs the optimal algorithm in Karp et al. [1990] on $A_1$, $A_2$, and so on. Suppose $\mathcal{R}$ assigns nodes to exactly $Y_k$ arrivals in $A_k$, then by the conclusion of Karp et al. [1990], we have $\mathbb{E}(Y_k) \geq (1 - 1/e)|A_k|$ where the expectation is taken over the random choice of $\sigma$. Hence, by lemma 2, the expected target value generated by $\sigma$-DUAL-RANKING is no less than the expected fake value of $\mathcal{R}$, so $\mathbb{E}\left(\sum_k d(k)Y_k\right) = \sum_k d(k)\mathbb{E}(Y_k) \geq \sum_k d(k)(1 - 1/e)|A_k| = (1 - 1/e)\sum_k d(k)|A_k|$. Note $\sum_k d(k)|A_k|$ is the optimal value, so we can conclude that RANKING has a competitive ratio of $1 - 1/e$ when $s = 1$.

Now consider the case where $s > 1$. We first fix an instance $\tau$ and construct a new instance $\tau'$ as follows. We split each arrival $i$ into $s$ arrivals $i_1, \ldots, i_s$. Moreover, for $i_1, \ldots, i_s$, we remove $j_{i,1}$ from the feasible set for arrival $i_1$, remove $j_{i,1}, j_{i,2}$ from the feasible set for arrival $i_2$, and so on. Specifically, for any $j$, we define $p_{i_k j}$ to be $p$ if $p_{ij} = p$ and $j \notin \{j_{i,1}, \ldots, j_{i,k-1}\}$, and 0 otherwise. In the following analysis, we restrict $s = 1$ whenever we talk about $\tau'$. Let $\text{ALG}_\tau$ and $\text{ALG}_{\tau'}$ respectively be the target values of RANKING when it runs on $\tau$ and $\tau'$ (note $s = 1$ when RANKING runs on $\tau'$), and let $\text{OPT}_\tau$ and $\text{OPT}_{\tau'}$ respectively be the optimal target values on $\tau$ and $\tau'$ (again, note $s = 1$ when $\tau'$ is analysed). We can see $\text{ALG}_\tau = \text{ALG}_{\tau'}$, and $\text{ALG}_{\tau'} \geq (1 - 1/e)\text{OPT}_{\tau'}$, where the inequality holds by our previous result for $s = 1$. Moreover, for any assignment of $\tau$, if nodes $j_1, \ldots, j_s$ are assigned to arrival $i$, we can arrange them properly to arrivals $i_1, \ldots, i_s$ in $\tau'$ without changing the target value, so $\text{OPT}_{\tau'} \geq \text{OPT}_\tau$. By combining the inequalities above, we have $\text{ALG}_\tau \geq (1 - 1/e)\text{OPT}_\tau$. This means RANKING also has a competitive ratio of $1 - 1/e$ when $s > 1$. $\square$

# 6 CONCLUSION

In this paper, we study a new variant of the online bipartite problem where each agent has a probability of rejecting an assignment. When a new agent arrives, previous assignment outcomes are not revealed in our setting. We give a deterministic algorithm with a tight competitive ratio of the problem. Next we propose an optimal randomized algorithm with a competitive ratio of $3/4$ when there are no more than two arrivals. We show that the competitive ratio can be $1 - 1/e$, which is tight, in a special case where the probabilities are either 0 or a fixed value $p$.

**Acknowledgements**

This work was partially supported by National Key Research and Development Program of China under (Grant No. 2020AAA0103401); National Natural Science Foundation of China (Grant No. 62172422); Beijing Outstanding Young Scientist Program (No. BJJWZYJH012019100020098); Intelligent Social Governance Interdisciplinary Platform, Major Innovation & Planning Interdisciplinary Platform for the "Double-First Class" Initiative, Renmin University of China.

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
