# OpenReview forum: "A Competitive Analysis of Online Failure-Aware Assignment"
_auai.org/UAI/2022/Conference — UAI 2022 Poster_

### Official Review · Reviewer_m7aC · 2022-04-07

**Q2(1) Originality/Novelty:** 3
**Q2(2) Significance/Impact:** 2
**Q2(3) Correctness/Technical Quality:** 3
**Q2(6) Clarity Of Writing:** 3
**Q6 Overall Score:** 5
**Q8 Confidence In Your Score:** 3

**Q1 Summary And Contributions:**

Thee paper studies a new version of the classical online bipartite matching problem where each agent has an unknown probability of rejecting an assignment. The problem is motivated by new advertisement applications in live streaming markets.

A 1/2 competitive deterministic algorithm is designed for this problem and it is proved that this competitive ratio is tight. Moreover, an optimal randomized algorithm is given for the case where each match is successful with probability p or 0.

**Q2 Assessment Of The Paper:**

More detailed information regarding each of these aspects is given below:

**Q2(5) Reproducibility:**

3: Good: Key resources (e.g., proofs, code, data) are available and key details (e.g., proofs, experimental setup) are sufficiently well-described for competent researchers to confidently reproduce the main results.

**Q3 Main Strengths:**

The paper deals with a variant of the classic online bipartite matching problem. This is a very-well studied problem that has a large impact on several areas of AI, in particular in the setting of online advertising and matching markets. The version of the problem proposed in the paper is motivated by specific applications in the setting of live streaming advertisement.

The paper appears to be technically solid and sound. Technical results are not trivial and proofs are given of all the main statements.
The organization of the paper is good.

**Q4 Main Weakness:**

No experiments are presented to support and complement the competitive analysis of the proposed algorithms. In particular, for the randomized algorithm it could help to have an experimental analysis of the competitiveness of the RANDOM algorithm in general (Theorem 3 holds only for the case with at most 2 arrivals and 1 node assigned).

Technical parts are not so easy to read and a little bit of intuition could help the reader that is not an expert in the field of online algorithms and competitive analysis.


**Q5 Detailed Comments To The Authors:**

Could you say something about the performance of algorithm RANDOM in more general cases?

**Q7 Justification For Your Score:**

The problem that is studied in the paper is relevant to the AI community and the presented results could have a moderate impact on the community of advertising on social media platforms.

The paper appears to be technically solid and sound. Results for the randomized case apply only to specific cases and a more general analysis is missing.

**Q9 Complying With Reviewing Instructions:**

1: Yes.

---

### Official Review · Reviewer_6W16 · 2022-04-12

**Q2(1) Originality/Novelty:** 3
**Q2(2) Significance/Impact:** 2
**Q2(3) Correctness/Technical Quality:** 3
**Q2(6) Clarity Of Writing:** 3
**Q6 Overall Score:** 6
**Q8 Confidence In Your Score:** 3

**Q1 Summary And Contributions:**

The authors study a new online matching problem with capacities.
They give a deterministic algorithm that has the best possible competitive factor (1/2). They also give a randomized algorithm that achieves a better competitive factor.

**Q2 Assessment Of The Paper:**

More detailed information regarding each of these aspects is given below:

**Q2(4) Quality Of Experiments (Optional):**

3: Good: The experimental evaluation is adequate, and the results convincingly support the main claims.

**Q2(5) Reproducibility:**

3: Good: Key resources (e.g., proofs, code, data) are available and key details (e.g., proofs, experimental setup) are sufficiently well-described for competent researchers to confidently reproduce the main results.

**Q3 Main Strengths:**

Tight results for the deterministic case.
Some exploration for the deterministic case.
Well-written
Rigorous analysis

**Q4 Main Weakness:**

The problem feels a little artificial. Why are all the s the same, but the c_i are different?

**Q5 Detailed Comments To The Authors:**

Make the x_i clearer on page 2

**Q7 Justification For Your Score:**

Seems like a nice paper. It won't revolutionize the field.

**Q9 Complying With Reviewing Instructions:**

1: Yes.

---

### Official Review · Reviewer_3ysH · 2022-04-12

**Q2(1) Originality/Novelty:** 3
**Q2(2) Significance/Impact:** 2
**Q2(3) Correctness/Technical Quality:** 3
**Q2(6) Clarity Of Writing:** 4
**Q6 Overall Score:** 7
**Q8 Confidence In Your Score:** 4

**Q1 Summary And Contributions:**

Motivated by the interaction between advertisers and influencers on social media platforms, the paper considers a variant of the online bipartite matching problem. .In addition, the paper considers a specific setting (specifying the characteristics of match between each influencer and advertiser). The paper proposes a deterministic and two randomized algorithms for the problem with guarantees on their competitive ratio, showing the optimality of the proposed randomized algorithm in some cases.

**Q2 Assessment Of The Paper:**

More detailed information regarding each of these aspects is given below:

**Q2(5) Reproducibility:**

3: Good: Key resources (e.g., proofs, code, data) are available and key details (e.g., proofs, experimental setup) are sufficiently well-described for competent researchers to confidently reproduce the main results.

**Q3 Main Strengths:**

1. The problem considered in the paper is both well motivated from an application perspective and interesting from a theoretical lens.
2. The proposed algorithms are appropriately described. Further, performance guarantees have been provided for these algorithms which go on to show optimality of the proposed algorithms in some cases.
3. The paper is well written. The organization of the paper makes it easy to follow the key ideas. Further, the related works section effectively communicates the existing literature and the similarities and differences of this paper with other relevant academic papers.


**Q4 Main Weakness:**

1. The section on the special case when each p_{ij} is either 0 or a fixed value p is not very well motivated in application. Intuitively, it seems that there would be large variations in fame of influencers, given that the audience outreach can vary significantly across influencers. It would be useful to elaborate further on the application motivation here.
2. The proof of Theorem 2, seems to be only for the case when n=2, s=1, c_1 = c_2 = 1. Proof of generalization to all cases has not been provided.
3. As currently stated, Theorem 2 is for all values of n, s, c whereas Theorem 3 seems to be only for n=2, s=1, c_1 = c_2 = 1. This distinction should be clearly stated in the text descriptions and the abstract.
4. For ease of following the proof arguments for Theorem 5, it would be helpful to have a proof sketch.
5. A numerical study on the performance of the proposed algorithms can further substantiate the value of the algorithms.


**Q5 Detailed Comments To The Authors:**

For improvement suggestions, please refer to the section on weaknesses.

**Q7 Justification For Your Score:**

The paper considers an interesting problem both from application and theoretical perspective. Given the increase in popularity of influencer advertising, the considered problem is very timely. The results in the paper build good knowledge on solutions for the considered problem. The paper is also well written.

**Q9 Complying With Reviewing Instructions:**

1: Yes.

---

### Decision · Program_Chairs · 2022-05-15

**Decision:**

Accept (Poster)

**Comment:**

Meta Review: This paper considers a variant of the online bipartite matching problem and proposes corresponding algorithms with guarantees on their competitive ratio. The reviewers are all positive about this paper. The paper could be more complete if some empirical analyses were provided.